# Multi-Modal Object Re-Identification via Sparse Mixture-of-Experts

Yingying Feng [* 1]  Jie Li [* 2]  Chi Xie [3]  Lei Tan [4]  Jiayi Ji [2 4]

## Abstract

We present MFRNet, a novel network for multi-modal object re-identification that integrates multi-modal data features to effectively retrieve specific objects across different modalities. Current methods suffer from two principal limitations: (1) insufficient interaction between pixel-level semantic features across modalities, and (2) difficulty in balancing modality-shared and modality-specific features within a unified architecture. To address these challenges, our network introduces two core components. First, the Feature Fusion Module (FFM) enables fine-grained pixel-level feature generation and flexible cross-modal interaction. Second, the Feature Representation Module (FRM) efficiently extracts and combines modality-specific and modality-shared features, achieving strong discriminative ability with minimal parameter overhead. Extensive experiments on three challenging public datasets (RGBNT201, RGBNT100, and MSVR310) demonstrate the superiority of our approach in terms of both accuracy and efficiency, with 8.4% mAP and 6.9% accuracy improved in RGBNT201 with negligible additional parameters. The code is available at https://github.com/stone96123/MFRNet.

## 1. Introduction

Object Re-Identification (ReID) focuses on retrieving specific object across non-overlapping camera views using known information. Traditionally, single-modal ReID relies heavily on RGB images, which face significant challenges under adverse conditions such as poor lighting, shadows, and low image resolutions (Li et al., 2020; Zhang et al.,

*Equal contribution  [1]School of Computer Science and Engineering, Northeastern University, Shenyang, China. [2]School of Informatics, Xiamen University, Xiamen, China. [3]Tongji University, Shanghai, China. [4]National University of Singapore, Singapore. Correspondence to: Lei Tan <lei.tan@nus.edu.sg>.

*Proceedings of the 42nd International Conference on Machine Learning*, Vancouver, Canada. PMLR 267, 2025. Copyright 2025 by the author(s).

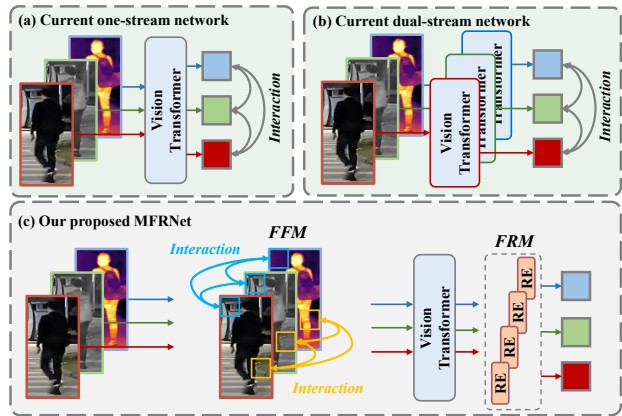

Figure 1. **Comparison between our proposed MFRNet and current mainstream structure.** (a) and (b) show the current mainstream one-stream and two-stream networks, respectively. Besides the limitation of neglecting the pixel-level alignment characteristics of multi-modal images, current methods also face the challenge of balancing the modality-specific and modality-shared representation in a unified network. (c) shows our proposed MFRNet and 'RE' refers to the representation expert. MFRNet inherits the idea of MoE and extends it with multi-modal fusion (FFM) and representation (FRM), allowing it to achieve fine-grained interaction and efficient representation for multi-modal data.

2021; Liu et al., 2024; Tan et al., 2024b). These limitations frequently result in the extraction of misleading features and a subsequent loss of discriminative information (Zheng et al., 2021; Wang et al., 2022). To address these challenges, multi-modal ReID (Shi et al., 2023; Lu et al., 2023; Shi et al., 2024; Yang et al., 2024) has emerged as a promising solution, leveraging complementary information from various modalities to enhance feature representation and achieve more acceptable identification ability in complex scenarios.

With the widespread application of Vision Transformer (ViT)(Alexey, 2020; Ji et al., 2025), numerous ViT-based ReID methods have emerged from general-purpose (He et al., 2021; Tan et al., 2022; 2024a) to multi-modal scenarios (Zhang et al., 2024a; Wang et al., 2024c). However, as illustrated in Figure 1, current mainstream methods face two fundamental limitations: insufficient interaction and feature imbalance. The insufficient interaction manifests in existing approaches that mainly emphasize interaction

at high-level semantic feature spaces (Wang et al., 2024a; Zhang et al., 2024a). These methods overlook the inherent pixel-to-pixel alignment characteristics in multi-modal images, resulting in inadequate fine-grained interaction between multi-modal information. The feature imbalance stems from the difficulty of balancing modality-shared and modality-specific features (Wang et al., 2022; Zhang et al., 2024a). A single parameter-sharing ViT backbone, though parameter-efficient, struggles to preserve modality-specific discriminative features. Alternatively, employing separate ViT backbones for different modalities can better preserve modal characteristics but introduce significant parameter redundancy and computational overhead.

To address the insufficient interaction and feature imbalance challenges, we propose a Modality Fusion and Representation Network (MFRNet). Our design incorporates mixture-of-experts (Jacobs et al., 1991) paradigm into both generators and feature extractors, achieving pixel-level cross-modal interaction and balanced feature learning. MFRNet consists of two key components: Feature Fusion Module (FFM) with mixture-of-generators for fine-grained interaction, and Feature Representation Module (FRM) with mixture of representation experts for balanced feature extraction. The FFM tackles insufficient interaction through its mixture-of-generators design. Inspired by RLE (Tan et al., 2024c) that cross-spectral transformation exhibits locally linear characteristics with surface-dependent variations, FFM employs multiple simple generators rather than complex unified structures. Through the mixture-of-generators mechanism, diverse tokens spontaneously select the most suitable generator for different modalities, locations, and attribute tokens. Additionally, FFM achieves fine-grained information exchange through weighted feature fusion, where fusion weights are learned via a generative network. The FRM addresses feature imbalance through a mixture of representation experts structure. It dynamically routes tokens from different modalities to various experts, enabling efficient parameter utilization while preserving both modality-shared and modality-specific features. Through dynamic expert activation, different experts in FRM adaptively focus on various shared visual attributes while maintaining necessary modality-specific representations. Notably, this design enables our framework to handle missing modalities without specific architectural modifications or training procedures.

The contributions of this paper are summarized as follows:

- We propose a Modality Fusion and Representation Network (MFRNet) for multi-modal object re-identification, which inherits the idea of a sparse mixture of experts and extends it with multi-modal fusion and representation.

- We introduce a Feature Fusion Module (FFM) and a Feature Representation Module (FRM). The former aims to achieve fine-grained interaction between multi-modal inputs, while the latter aims to achieve efficient and balanced feature extraction between modality-shared and modality-specific representations.

- Extensive experiments on three public multi-modal object ReID datasets including RGBNT201, RGBNT100, and MSVR310, verifying the superior performance of MFRNet.

## 2. Related Work

### 2.1. Multi-modal Person ReID

Human-centric computer vision tasks (Shen & Tang, 2024; Shen et al., 2025) have long been a central focus within the research community. Unlike single-modal object re-identification (ReID), multi-modal person ReID includes more information and is suitable for a wider range of applications. Existing methods (Park et al., 2021; Zheng et al., 2023; 2022; He et al., 2023; Tan et al., 2023; Zhang & Wang, 2023) for multi-modal object ReID mainly focus on how to effectively integrate information from multiple modalities and alleviate the cross-modal heterogeneous issue. For example, Zheng et al. (Zheng et al., 2021) utilize the complementary advantages of multiple modalities to propose PFNet for learning effective multi-modal features. To boost modality-specific representations, Wang et al. (Wang et al., 2022) proposed the IEEE method, which includes important information exchange, feature enrichment, and intra-class discrepancy maximization mechanisms. To effectively utilize the relationship of modalities to reduce the gap between different modalities, Guo et al. (Guo et al., 2022) proposed the GAFNet model to fuse multiple data sources. Additionally, the vision transformers (ViT) (Alexey, 2020; Radford et al., 2021; Pan et al., 2022; 2023; Crawford et al., 2023; Wang et al., 2024a) model has achieved good results in many fields, promoting the development of ViT-based multi-modal person ReID algorithms. Wang et al. (Wang et al., 2024b) explored the influence of global and local features of ViT and proposed the GLTrans model. Wang et al. (Wang et al., 2024c) explored the domain traits of unlabeled test data and proposed a heterogeneous test-time training framework based on ViT to improve generalization performance. Yu et al. (Yu et al., 2024) proposed the RSCNet model based on the idea of token sparsification in ViT to alleviate the multi-modal heterogeneity problem. Zhang et al. (Zhang et al., 2024a) proposed the EDITOR model, which selects diverse tokens from ViT to mitigate the effect of irrelevant backgrounds and reduce the gap between modalities.

Despite the good performance of these methods, they tend to employ high-level semantic multi-modal feature fusion, which overlooks the fine-grained spatial alignment characteristics of multi-modal data. Additionally, they struggle

to efficiently extract modality-shared and modality-specific multi-modal representation within a unified network. Therefore, we propose MFRNet, which adopts a mixture of generator experts to leverage the pixel-to-pixel alignment of multi-modal data and introduces a mixture of representation experts to adaptively extract modality-shared features while retaining modality-specific features.

## 2.2. Mixtures of Experts

The Mixture of Experts (MoE) (Jacobs et al., 1991) originates from Ensemble Learning (Ganaie et al., 2022) and can effectively enhance model performance. Notably, after Lepikhin et al. (Lepikhin et al., 2020) combined MoE with the transformer framework and proposed the Gshard model, it garnered significant attention across various fields, leading to the development of more models incorporating MoE technology (Chen & Wang, 2025; Dai et al., 2024; Gui et al., 2024; Lin et al., 2024; Liu et al., 2025; Mustafa et al., 2022; Li et al., 2025). Subsequently, to facilitate the implementation and flexible design of MoE, Hwang et al. (Hwang et al., 2023) open-sourced the Tutel codebase. Additionally, to reduce training and inference costs, Chowdhury et al. (Chowdhury et al., 2023) proposed pMoE, which can filter label-irrelevant patches and route similar class-discriminative patches to the same expert. Recently, many works have attempted to introduce MoE into ReID. Among them, Li et al. (Li et al., 2023) proposed the MPC model, which leverages MoE to learn multiple pseudo-label spaces, thereby flexibly handling different variations. Xu et al. (Xu et al., 2022) proposed the META model based on MoE to better exploit the domain-invariant characteristics of multi-modal data. Kuang et al. (Kuang et al., 2024) proposed the MiKeCoCo model, which incorporates multiple experts with unique perspectives into CLIP and fully leverages high-level semantic knowledge for comprehensive feature representation. Additionally, Li et al. (Li et al., 2022) explored an orthogonal direction for domain generalization research and proposed the GMoE model by combining ViT and MoE technologies. Based on the results of the aforementioned studies, we leveraged the advantages of MoE technology and utilized the Tutel framework to design and implement MoE techniques to enhance model performance.

## 3. Methodology

As shown in Figure 2, the proposed MFRNet employs a CLIP pre-trained vanilla ViT-base as its basic backbone network. On this basis, our network comprises two principal components, *i.e.*, the feature fusion module (FFM) and the feature representation module (FRM). The FFM introduces the pixel-level feature generation for flexible cross-modal interaction, which is also suitable for modal-missing scenarios, while the FRM inherits the idea of the sparse

Mixture-of-Experts (MoE) to adapt to diverse modal input at an extremely low cost and performs joint optimization of modality-specific and modality-shared features.

## 3.1. Feature Fusion Module

In general, a given image can be divided into patches of the same size and mapped to a fixed feature dimension through patchify operations. Then, we have the image feature as:

$$I_M = \{[I_M^{cls}], I_M^1, I_M^2, \cdots, I_M^n\} \quad \text{where } M \in R, N, T, \tag{1}$$

where R, N, T denote RGB, NIR, and TIR modalities respectively. Given that multi-modal images maintain pixel-level alignment, our feature fusion module (FFM) is designed to enable effective interaction and mutual enhancement between different modality features. As illustrated by the RLE (Tan et al., 2024c) and AMML (Zhang et al., 2024b), the cross-spectral transformation can be considered as a local linear transformation, while the linear factor will be varied across different surfaces. It shows that the transformation between different spectra is quite simple but variant. Therefore, inspired by the idea of the Mixture-of-experts strategy and the above observation, FFM attempts to combine multiple simple generators to adapt to diverse input tokens. In addition, it is interesting to find that this kind of fusion strategy of FFM also works well for modality-missing scenarios in a modality-complementing way.

Specifically, as shown in Figure 2, taking the RGB feature $I_R$ for example, it can be re-generated with NIR and TIR features as:

$$I_R^g = \lambda \times I_R + w_R^N(I_N) \times g_R^N(I_N) + w_R^T(I_T) \times g_R^T(I_T), \tag{2}$$

where $\lambda$ is hyper-parameters to determine the proportion of information retained by the modality itself, $w_R^N(\cdot)$ and $w_R^T(\cdot)$ generate the weighting coefficients for the features generated from NIR and TIR, respectively. And $g_R^N(\cdot)$ and $g_R^T(\cdot)$ represent the reconstructed RGB features from the NIR and TIR modality, respectively. The re-generated features of NIR and TIR are similar.

For the weight coefficient function $w(\cdot)$[1], we define it as follows:

$$w(I) = fc\big(AvgPool(I)\big), R^{n \times d} \Rightarrow R^{1 \times d} \Rightarrow R^{1 \times 1}, \tag{3}$$

where $I \in R^{n \times d}$ represents image feature from specific modality, $AvgPool$ represents global average pooling, and $fc$ represents a linear layer. To normalize the weight coefficients, we apply the Softmax operation on the weights of each reconstructed modality feature like

---

[1]We omit the subscript for modality for simplification.

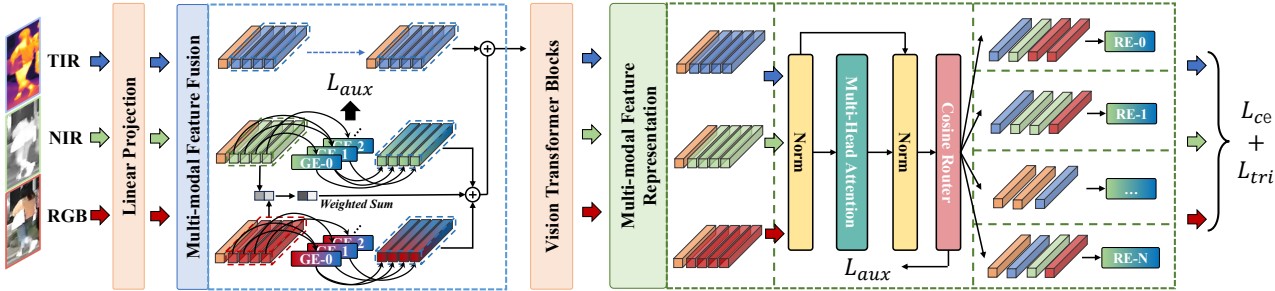

*Figure 2.* **Overall architecture of MFRNet.** MFRNet is built from the basic Transformer blocks inherited from vanilla ViT, with two new modules (feature fusion module and feature representation Module) added to adapt to multi-modal object re-identification tasks. In the blue box, the Feature Fusion Module (FFM) employs multiple simple generators to adaptively provide fine-grained interaction information, while in the green box, the Feature Representation Module (FRM) employs diverse representation experts to extract and combine modality-specific and modality-shared features. Here, GE refers to the generation experts in FFM, while RE represents the representation experts in FRM. Notably, the figure illustrates the NIR+RGB → TIR interaction, with the same interaction applying to NIR and RGB.

$Softmax\left(w_R^N(I_N), w_R^T(I_T)\right)$. The weight generation function is modality-specific, and there are total 6 functions for 3 modalities.

For the feature generation function $g(\cdot)$, as mentioned before, the transformation between different modality images is simple but variant. Compared to complex structures, simple generators work well (Zhang et al., 2024b). Therefore, we apply the mixture-of-experts (MoE) architecture (Li et al., 2022) to combine multiple simple generators. We define the simplest generation expert $g_f(\cdot)$ as:

$$g_f(I) = conv_2\left(drop\left(conv_1(I)\right)\right). \tag{4}$$

Then we enhance it using sparse MoE technology. Specifically, we categorize feature generation into two types based on the different modalities: $\{I_{N\to R}, I_{T\to R}, I_{R\to N}\}$ and $\{I_{T\to N}, I_{R\to T}, I_{N\to T}\}$. Since each of the three modalities can generate features for the other two modalities, we define two moe layers. Finally, the feature generation function is as follows:

$$I_{experts} = \{g_f\left(fc_i(I)\right) \mid \text{for } i \in [1, K]\}$$
$$S_{router} = Softmax\left(\frac{W_1^F I W_e^F}{\tau \left\|W_1^F I\right\| \left\|W_e^F\right\|}\right)$$
$$Gate(X) = Top_k(S_{router}) \tag{5}$$
$$g(I) = Gate(I) \cdot I_{experts},$$

where $fc_i$ refers to the linear layer for $u_{th}$ expert, $S_{router}$ represents cosine router which indicates routing scores for selecting experts. $\|\cdot\|$ indicate the $L_2$ normalization, and $\tau$ indicates the learnable temperature scaler. $W_1^F$ is the weight of linear projection which aims to project the input token to a low-rank subspace, while $W_e^F$ is the expert embedding that transforms the feature to final scoring distribution. Note that $k = 1$ represents only the highest-scoring expert is selected for feature generation each time.

As stated before, although do not have any specific training, we find that FFM can also work well for modality-missing

scenarios in a modality-complementing way. Specifically, for the modality missing scenarios during testing, we develop conditional reconstructed rules for modality $M$ as:

$$I_M^g = \begin{cases} \lambda I_M + \sum_{S\in\mathcal{A}} w_M^S(I_S) \times g(I_S) & \text{if } M \text{ present} \\ \sum_{S\in\mathcal{A}} w_M^S(I_s) \times g(I_s) & \text{if } M \text{ missing,} \end{cases} \tag{6}$$

where $\mathcal{A}$ denotes available modalities. For example, if RGB is missing, its features are generated based on NIR and TIR as follows: $I_R = w_R^N(I_N) \times g(I_N) + w_R^T(I_T) \times g(I_T)$. Here, $g(I_N)$ denotes the features for RGB generated from NIR, $g(I_T)$ represents the features for RGB generated from TIR, $w_R^N(I_N)$ and $w_R^T(I_T)$ are the respective weights.

### 3.2. Feature Representation Module

Existing approaches face fundamental challenges in balancing modality-shared and modality-specific information. While employing separate networks for each modality preserves modality-specific characteristics, this introduces significant structural and computational redundancy. Conversely, using a fully shared single network for all modalities exclusively captures shared representations across modalities. Hence, we propose the Feature Representation Module (FRM), which integrates a Mixture-of-Experts architecture to preserve valuable modality-specific information with parameter-efficient implementation.

With the reconstructed multi-modal image features $I^g = (I_R^g, I_N^g, I_T^g)$, we use a vision transformer (ViT) to extract fundamental features. Within this module, we aim to capture modality-specific feature representations within a unified architectural framework.

Specifically, we modify the residual attention layer based on RepAdapter as follows: To extract unique features for different modality data, we employ RepAdapter (Luo et al., 2023), which is designed to rapidly adapts to vision tasks by inserting adapter layers between key layers. Specifically,

the residual attention layer can be represented as:

$$Y = X + \text{MH-Attn}(\text{LayerNorm}(X)),$$
$$Z = Y + R_{Ada}(\text{LayerNorm}(Y)), \quad (7)$$

where $X$ and $Z$ are the input and the output of the current layer, MH-Attn$(\cdot)$ represents the multi-head attention layer, $R_{Ada}(\cdot)$ represents the RepAdapter layer. The formula of the RepAdapter layer is as:

$$R_{Ada}(Y_{lnorm}) = Y_{lnorm} + conv_B(drop(conv_A(Y_{lnorm}))), \quad (8)$$

where $Y_{lnorm}$ refers to the $Y$ after the LayerNorm layer, $conv_A$ and $conv_B$ represent 1D convolutions with kernel size of $1 \times 1$ and groups set to 1. Similarly, to better adapt to multi-modal data, we use $R_{Ada}$ as the expert and enhance it by the MoE (Li et al., 2022) layer.

$$S_{router}^{Ada} = Softmax(\frac{W_1^R Y_{lnorm} W_e^R}{\tau \|W_1^R Y_{lnorm}\| \|W_e^R\|}),$$
$$Gate_{Ada}(Y_{lnorm}) = Top_k(S_{router}^{Ada}), \quad (9)$$

where $\tau = 0.01$ and $k = 1$. It is important to note that since the number of modalities is 3, the number of experts should be greater than 3. $W_1^R$ is the weight of the linear projection, which aims to project the input token to a low-rank subspace, while $W_e^R$ is the expert embedding that transforms the feature to the final scoring distribution.

Based on the above derivation, using $l_{Ada}^{moe}$ to represent the MoE-enhanced RepAdapter, the modality-specific features are as follows:

$$Z = Y + l_{Ada}^{moe}(Y_{lnorm}). \quad (10)$$

Here, the auxiliary loss for MoE is denoted as $L_{aux}^{Ada}$.

### 3.3. Objective Function

As shown in Figure 2, our model consists of three main loss functions: losses for the ViT backbone, auxiliary losses for the multi-modal feature completion module, and auxiliary losses for the modality-specific feature representation module. For the losses for the ViT backbone, we follow previous research by using label smoothing cross-entropy loss (Szegedy et al., 2016) and triplet loss (Hermans et al., 2017) for optimizing the representation. Besides the basic losses, following the (Riquelme et al., 2021; Li et al., 2022), we define two auxiliary losses: important loss $\mathcal{L}_{imp}$ and load loss $\mathcal{L}_{load}$ to balance the frequency being activated for the expert. The important loss $\mathcal{L}_{imp}$ to encourage a balanced usage of different experts across tokens and load loss $\mathcal{L}_{load}$ to encourage balanced assignment across experts. Specifically, the importance of expert $e$ in a batch of images $I_b$ is

defined as the normalized routing weight corresponding to expert $i$ summed over images as:

$$\text{Imp}_e(I_b) = \sum_{I \in I_b} S_{router}^e. \quad (11)$$

The important loss $\mathcal{L}_{imp}$ is given by the squared coefficient of variation of the importance distribution over experts as:

$$\mathcal{L}_{imp}(I_b) = (\frac{STD(Imp(I_b))}{MEAN(Imp(I_b))})^2. \quad (12)$$

Meanwhile, the load loss $\mathcal{L}_{load}$ can be given as:

$$\mathcal{L}_{load}(I_b) = \frac{STD(load(I))}{MEAN(load(I))},$$
$$with \quad load_e(I_b) = \sum_{I \in I_b} p_e(I),$$
$$and \quad p_e(I) = 1 - \Phi(\frac{\eta_k - (W_1 I W_e)_e}{\sigma}). \quad (13)$$

Herein, the $\eta_k$ refers to the $K_{th}$ largest entry after softmax while $\Phi$ indicates the cumulative distribution function of a Gaussian distribution. Therefore, the whole auxiliary losses $\mathcal{L}_{aux}$ is formulated as:

$$\mathcal{L}_{aux} = \frac{\mathcal{L}_{imp} + \mathcal{L}_{load}}{2}. \quad (14)$$

Finally, the total loss function of MFRNet is given as follows:

$$\mathcal{L} = \mathcal{L}_{ce}^{ViT} + \mathcal{L}_{tri}^{ViT} + (\mathcal{L}_{aux}^{moe1} + \mathcal{L}_{aux}^{moe2}) + \mathcal{L}_{aux}^{Ada}. \quad (15)$$

## 4. Experiments

### 4.1. Implementation

**Datasets.** We evaluate our model performance on three public multi-modal object ReID datasets. Specifically, RGBNT201 (Zheng et al., 2021) is a multi-modal person ReID dataset, which includes 4,787 aligned RGB, NIR, and TIR images from 201 identities. RGBNT100 (Li et al., 2020) is a large-scale multi-modal vehicle ReID dataset comprising 17,250 image triples. It covers a wide range of challenging visual conditions, making it suitable for assessing the robustness of vehicle ReID methods. MSVR310 (Zheng et al., 2022) is a small-scale multi-modal vehicle ReID dataset that includes 2,087 high-quality image triples captured across diverse environments and time spans, providing complex visual challenges for vehicle ReID evaluation.

**Evaluation protocols.** To assess the performance of our method, we utilize the mean Average Precision (mAP) and Cumulative Matching Characteristics (CMC) at Rank-K ($K = 1, 5, 10$). These metrics are standard in this field and provide a comprehensive evaluation of our model's effectiveness. Additionally, we present the number of parameters and FLOPs to analyze the complexity of our model.

*Table 1.* **Performance comparison on RGBNT201.** The best and second results are in bold and underlined, respectively.

| | Methods | *m*AP | R-1 | R-5 | R-10 |
|---|---|---|---|---|---|
| **Single** | MUDeep (Qian et al., 2017) | 23.8 | 19.7 | 33.1 | 44.3 |
| | HACNN (Li et al., 2018) | 21.3 | 19.0 | 34.1 | 42.8 |
| | MLFN (Chang et al., 2018) | 26.1 | 24.2 | 35.9 | 44.1 |
| | PCB (Sun et al., 2018) | 32.8 | 28.1 | 37.4 | 46.9 |
| | OSNet (Zhou et al., 2019) | 25.4 | 22.3 | 35.1 | 44.7 |
| | CAL (Rao et al., 2021) | 27.6 | 24.3 | 36.5 | 45.7 |
| **Multi** | HAMNet (Li et al., 2020) | 27.7 | 26.3 | 41.5 | 51.7 |
| | PFNet (Zheng et al., 2021) | 38.5 | 38.9 | 52.0 | 58.4 |
| | IEEE (Wang et al., 2022) | 47.5 | 44.4 | 57.1 | 63.6 |
| | DENet (Zheng et al., 2023) | 42.4 | 42.2 | 55.3 | 64.5 |
| | UniCat (Crawford et al., 2023) | 57.0 | 55.7 | - | - |
| | HTT (Wang et al., 2024c) | 71.1 | 73.4 | 83.1 | 87.3 |
| | EDITOR (Zhang et al., 2024a) | 66.5 | 68.3 | 81.1 | 88.2 |
| | RSCNet (Yu et al., 2024) | 68.2 | 72.5 | - | - |
| | TOP-ReID (Wang et al., 2024a) | 72.3 | 76.6 | 84.7 | 89.4 |
| | **Ours** | **80.7** | **83.6** | **91.9** | **94.1** |

**Implementation details.** Our model is implemented using the PyTorch toolbox, and experiments are conducted on an NVIDIA V100 GPU. We utilize pre-trained CLIP (Radford et al., 2021) as the visual encoder. Images are resized to $256 \times 128$ for RGBNT201 and $128 \times 256$ for RGBNT100 and MSVR310. For data augmentation, we apply random horizontal flipping, cropping, and erasing following (Zhong et al., 2020). The mini-batch size is set to 128 for RGBNT100, and 64 for RGBNT201 and MSVR310, with corresponding sampling strategies for each dataset. We employ the Adam optimizer with an initial learning rate of $3.5e-4$ and the learning rate of the visual encoder is $5e-6$. The total number of training epochs is set to 45 for RGBNT201 and RGBNT100, and 50 for MSVR310.

### 4.2. Comparison with State-of-the-Art Methods

**multi-modal Person ReID.** We compare the proposed MFRNet with several single-spectral and multi-modal methods on the RGBNT201 dataset, as shown in Table 1. It is evident that multi-modal methods generally outperform single-spectral methods by leveraging complementary multi-modal information. Among current multi-modal methods, TOP-ReID achieves the best performance, with 72.3% mAP, 76.6% R-1, 84.7% R-5, and 89.4% R-10. Our proposed method improves these metrics by 8.4%, 6.9%, 7.2%, and 4.7%, which demonstrates the superior performance of MFRNet for multi-modal person ReID. Notably, we use the CLIP-based ViT to reproduce TOP-ReID, and the results show that our method surpasses TOP-ReID by approximately 9.7% mAP and 9.6% R-1.

**multi-modal Vehicle ReID.** Table 2 shows the comparison

*Table 2.* **Performance comparison on RGBNT100 and MSVR310.** The best and second results are in bold and underlined, respectively.

| | Methods | RGBNT100 | | MSVR310 | |
|---|---|---|---|---|---|
| | | *m*AP | R-1 | *m*AP | R-1 |
| **Single** | PCB (Sun et al., 2018) | 57.2 | 83.5 | 23.2 | 42.9 |
| | MGN (Wang et al., 2018) | 58.1 | 83.1 | 26.2 | 44.3 |
| | DMML (Chen et al., 2019) | 58.5 | 82.0 | 19.1 | 31.1 |
| | BoT (Luo et al., 2019) | 78.0 | 95.1 | 23.5 | 38.4 |
| | OSNet (Zhou et al., 2019) | 75.0 | 95.6 | 28.7 | 44.8 |
| | Circle Loss (Sun et al., 2020) | 59.4 | 81.7 | 22.7 | 34.2 |
| | HRCN (Zhao et al., 2021) | 67.1 | 91.8 | 23.4 | 44.2 |
| | TransReID (He et al., 2021) | 75.6 | 92.9 | 18.4 | 29.6 |
| | AGW (Ye et al., 2022) | 73.1 | 92.7 | 28.9 | 46.9 |
| **Multi** | HAMNet (Li et al., 2020) | 74.5 | 93.3 | 27.1 | 42.3 |
| | PFNet (Zheng et al., 2021) | 68.1 | 94.1 | 23.5 | 37.4 |
| | GAFNet (Guo et al., 2022) | 74.4 | 93.4 | - | - |
| | GraFT (Yin et al., 2023) | 76.6 | 94.3 | - | - |
| | GPFNet (He et al., 2023) | 75.0 | 94.5 | - | - |
| | PHT (Pan et al., 2023) | 79.9 | 92.7 | - | - |
| | UniCat (Crawford et al., 2023) | 79.4 | 96.2 | - | - |
| | CCNet (Zheng et al., 2022) | 77.2 | 96.3 | 36.4 | 55.2 |
| | HTT (Wang et al., 2024c) | 75.7 | 92.6 | - | - |
| | TOP-ReID (Wang et al., 2024a) | 81.2 | 96.4 | 35.9 | 44.6 |
| | EDITOR (Zhang et al., 2024a) | 82.1 | 96.4 | 39.0 | 49.3 |
| | RSCNet (Yu et al., 2024) | 82.3 | 96.6 | 39.5 | 49.6 |
| | **Ours** | **88.2** | **97.4** | **50.6** | **64.8** |

results of our proposed MFRNet with other methods on the RGBNT100 and MSVR310 datasets. For the RGBNT100 dataset, TransReID achieves the best performance among single-spectral methods, while RSCNet significantly outperforms it among multi-modal methods. However, for the MSVR310 dataset, AGW achieves the best performance among single-spectral methods, while RSCNet has the highest mAP and CCNet has the highest R-1 among multi-modal methods. Notably, our proposed method achieves significant improvements on both datasets, increasing mAP by 5.9% and R-1 by 0.8% on RGBNT100, and increasing mAP by 11.1% and R-1 by 9.6% on MSVR310. This demonstrates the superior performance of MFRNet for multi-modal vehicle re-identification.

**Evaluation on Missing-spectral Scenarios.** In real-world application scenarios, persistent modality availability cannot be guaranteed. Therefore, exploring the performance of models under the condition of missing modalities helps to explore the application boundaries of the model. Therefore, we also evaluate the performance of MFRNet under different modality-missing scenarios on the RGBNT201 dataset. As shown in Table 3, PCB achieves the highest average mAP and R-1 among single-spectral methods. TOP-ReID

*Table 3.* **Performance of missing-modality settings on RGBNT201.** "M (X)" means missing the X image modality. The best and second results are in bold and underlined, respectively.

| | Methods | M (RGB) | | M (NIR) | | M (TIR) | | M (RGB+NIR) | | M (RGB+TIR) | | M (NIR+TIR) | | Average | |
|---|---|---|---|---|---|---|---|---|---|---|---|---|---|---|---|
| | | *m*AP | R-1 | *m*AP | R-1 | *m*AP | R-1 | *m*AP | R-1 | *m*AP | R-1 | *m*AP | R-1 | *m*AP | R-1 |
| Single | MUDeep (Qian et al., 2017) | 19.2 | 16.4 | 20.0 | 17.2 | 18.4 | 14.2 | 13.7 | 11.8 | 11.5 | 6.5 | 12.7 | 8.5 | 15.9 | 12.9 |
| | HACNN (Li et al., 2018) | 12.5 | 11.1 | 20.5 | 19.4 | 16.7 | 13.3 | 9.2 | 6.2 | 6.3 | 2.2 | 14.8 | 12.0 | 13.3 | 10.7 |
| | MLFN (Chang et al., 2018) | 20.2 | 18.9 | 21.1 | 19.7 | 17.6 | 11.1 | 13.2 | 12.1 | 8.3 | 3.5 | 13.1 | 9.1 | 15.6 | 12.4 |
| | PCB (Sun et al., 2018) | 23.6 | 24.2 | 24.4 | 25.1 | 19.9 | 14.7 | 20.6 | 23.6 | 11.0 | 6.8 | 18.6 | 14.4 | 19.7 | 18.1 |
| | OSNet (Zhou et al., 2019) | 19.8 | 17.3 | 21.0 | 19.0 | 18.7 | 14.6 | 12.3 | 10.9 | 9.4 | 5.4 | 13.0 | 10.2 | 15.7 | 12.9 |
| Multi | PFNet (Zheng et al., 2021) | - | - | 31.9 | 29.8 | 25.5 | 25.8 | - | - | - | - | 26.4 | 23.4 | - | - |
| | DENet (Zheng et al., 2023) | - | - | 35.4 | 36.8 | 33.0 | 35.4 | - | - | - | - | 32.4 | 29.2 | - | - |
| | TOP-ReID (Wang et al., 2024a) | 54.4 | 57.5 | 64.3 | 67.6 | **51.9** | **54.5** | 35.3 | 35.4 | 26.2 | 26.0 | 34.1 | 31.7 | 44.4 | 45.4 |
| | **Ours** | **64.7** | **65.2** | **72.3** | **76.1** | 51.6 | 49.5 | **41.4** | **43.4** | **27.3** | **27.9** | **37.2** | **35.6** | **49.1** | **49.6** |

*Table 4.* **Comparison of computational cost with recent methods.** We show the best result in bold.

| Methods | Params(M) | Flops(G) |
|---|---|---|
| HTT (Wang et al., 2024c) | 85.6 | 33.1 |
| EDITOR (Zhang et al., 2024a) | 117.5 | 38.6 |
| TOP-ReID (Wang et al., 2024a) | 278.2 | 34.5 |
| Ours | **57.1** | **22.1** |

*Table 5.* **Comparison with different modules.** We show the best score in bold.

| Index | Modules | | Metrics | | | | Params | FLOPs |
|---|---|---|---|---|---|---|---|---|
| | FFM | FRM | *m*AP | R-1 | R-5 | R-10 | M | G |
| 1 | ✗ | ✗ | 69.2 | 76.3 | 84.8 | 89.5 | 57.2 | 22.1 |
| 2 | ✓ | ✗ | 75.1 | 78.1 | 87.4 | 92.0 | 60.8 | 23.5 |
| 3 | ✗ | ✓ | 77.8 | 80.9 | 88.6 | 92.2 | **53.5** | **20.7** |
| 4 | ✓ | ✓ | **80.7** | **83.5** | **91.9** | **94.1** | 57.1 | 22.1 |

*Table 6.* **Performance analysis under different numbers of experts for FRM.** We show the best score in bold.

| Number | *m*AP | R-1 | R-5 | R-10 | Average |
|---|---|---|---|---|---|
| 3 | 77.6 | 78.6 | 87.9 | **92.3** | 84.1 |
| 6 | **77.8** | **80.9** | 88.6 | 92.2 | **84.9** |
| 9 | 75.1 | 76.7 | 87.8 | **92.3** | 83.0 |

*Table 7.* **Performance analysis of FRM at different locations within ViT.** We show the best score in bold.

| Layer | *m*AP | R-1 | R-5 | R-10 | Average |
|---|---|---|---|---|---|
| 10,11,12 | 71.6 | 75.5 | 87.1 | 90.8 | 81.2 |
| 10,12 | 75.7 | 77.6 | 87.6 | 91.7 | 83.1 |
| 12 | **77.8** | **80.9** | **88.6** | **92.2** | **84.9** |

outperforms it with an average performance of 44.4% mAP and 45.4% R-1 among multi-modal methods. Our proposed method improves these metrics by 4.7% mAP and 4.2% R-1 over TOP-ReID, validating the adaptability of our model to modality missing scenarios. It is worth noting that compared to the TOP-ReID which has a specific training phase for the modality missing condition, our MFRNet framework notably trains without explicit task-specific architectural modifications or dedicated training strategies, yet shows its intrinsic capability to compensate for the missing modality.

### 4.3. Ablation Study

To explore the effectiveness of different components in our model, we conduct comprehensive ablation studies on the RGBNT201 dataset, as shown in Table 5.

**Feature Fusion Module.** We use a shared vision transformer combined with label smoothing cross-entropy and triplet loss as our base framework. Building on this, we integrate the Feature Fusion Module (FFM), which boosts the performance on the mAP, R-1, R-5, and R-10 metrics by 5.9%, 1.8%, 2.6%, and 2.5% respectively. This shows that FFM effectively enhances overall model performance by generating and interacting multi-modal features using the fine-grained alignment of multi-modal data. However,

integrating the FFM module increases the model's parameters and FLOPs by 3.6M and 1.4G, respectively, therefore further efforts are needed to reduce them.

**Feature Representation Module.** Building on the aforementioned base framework, we introduce the Feature Representation Module (FRM). Experimental results show that integrating the FRM module into the baseline model improves mAP, R-1, R-5, and R-10 by 8.6%, 4.6%, 3.8%, and 2.7%, respectively. Notably, the model's parameters and FLOPs decrease by 3.7M and 1.4G as the FRM module uses RepAdapter to build MoE representations, reducing computational cost with two convolutional layers. This demonstrates that the FRM module enhances model accuracy by effectively representing shared and specific features of multi-modal data while using fewer parameters.

**MFRNet.** MFRNet, which integrates the FFM and FRM modules, achieves 80.7% mAP, 83.5% R-1, 91.9% R-5, and 94.1% R-10. In addition, MFRNet reduces the parameters by 0.1M compared to the baseline framework, while being on par with the baseline model on FLOPS. Consequently, the design of effective multi-modal feature fusion and representation mechanisms enables more precise multi-modal object re-identification.

*Table 8.* **Performance analysis under different experts number for FFM.** We show the best score in bold.

| Number | *m*AP | R-1 | R-5 | R-10 | Average |
|--------|-------|------|------|------|---------|
| 1 | 78.7 | 81.0 | 90.8 | 93.5 | 86.0 |
| 2 | 74.9 | 78.6 | 86.8 | 90.8 | 82.8 |
| 3 | **80.7** | **83.5** | **91.9** | **94.1** | **87.5** |
| 6 | 76.9 | 80.4 | 88.2 | 90.6 | 84.0 |
| 9 | 79.2 | 82.3 | 90.7 | 93.5 | 86.4 |
| 10 | 76.4 | 80.1 | 88.0 | 92.0 | 84.1 |
| 11 | 74.1 | 78.9 | 87.2 | 92.2 | 83.1 |
| 12 | 74.1 | 77.8 | 86.8 | 91.1 | 82.4 |

### 4.4. More Analysis

**Analysis of model computational complexity.** We compared computational costs across three major recent works. As shown in Table 4, our method achieves the lowest Params and FLOPs, measuring 57.1M and 22.1G, respectively.

**Experts Number in FRM.** We explore the model's performance with different numbers of experts in the FRM, where experts are responsible for extracting modality-specific features. As shown in Table 6, the performance is optimal when the number of experts is 6. Although more experts may make sense in maintaining modality-specific representations, this may also limit the network's learning of modality-shared knowledge.

**The Location of FRM.** We further explore the model's performance when the FRM module is placed in different layers of the ViT. As shown in Table 7, using the FRM only in the final layer is sufficient to capture modality-specific features for multi-modal data, achieving optimal performance.

**Expert visualization of FRM.** To visually observe the allocation of multiple experts in the FRM when extracting multi-modal features, we conduct visualizations on the RGBNT201 test set. The numbers in Figure 3 represent the index of the six experts. We also utilize similar colors to mark patches that are focused on by the same expert. It can be observed that Expert 0 concentrates on extracting pedestrian features in the NIR modality, while Experts 1, 2, and 3 focus on pedestrian regions in the RGB and TIR modalities. Additionally, Experts 4 and 5 tend to extract background features across all modalities. As our initial hypothesis predicted, semantically similar content achieves knowledge sharing through selecting analogous experts, while modality-specific representations that resist fusion maintain their distinct characteristics via dedicated expert allocation.

**Experts Number in FFM.** Similarly, we analyze the model's performance with different numbers of experts in the FFM, where experts are responsible for generating cross-

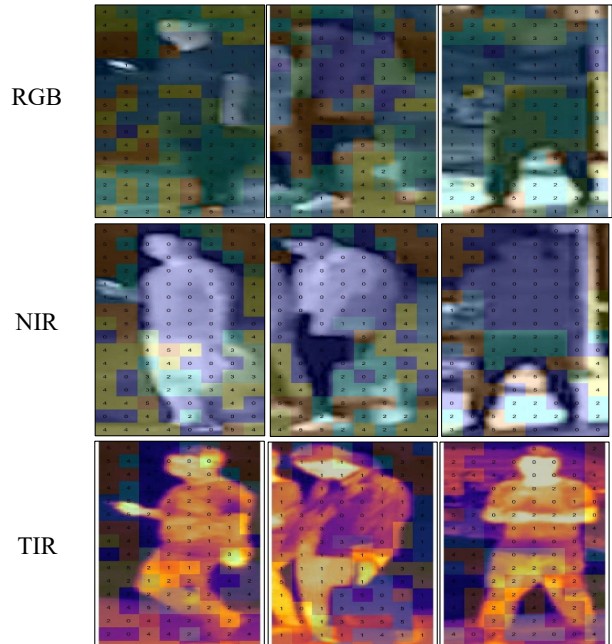

*Figure 3.* **Visualization of expert in FRM.** As expected earlier, similar content can share their knowledge by selecting similar experts to reduce redundancy, while for modality-specific representations that are difficult to fuse, their knowledge can also be retained by new experts.

*Table 9.* **Performance analysis under different $\lambda$ for FFM.** We show the best score in bold.

| Scale of $\lambda$ | *m*AP | R-1 | R-5 | R-10 | Average |
|--------------------|-------|------|------|------|---------|
| 0.0 | 4.7 | 2.4 | 9.8 | 18.4 | 8.8 |
| 0.3 | 79.9 | 82.1 | 89.2 | 92.8 | 86.0 |
| 0.5 | **80.7** | **83.5** | **91.9** | **94.1** | **87.5** |
| 0.7 | 74.8 | 77.8 | 86.1 | 90.4 | 82.3 |
| 1.0 | 77.5 | 81.5 | 88.8 | 92.5 | 85.1 |

modal features. As shown in Table 8, the model achieves the highest average performance when the number is 3.

**Parameter $\lambda$ of FFM.** For the parameter $\lambda$ in the FFM, we analyze the model's performance at different values. As shown in Table 9, the model performs the worst when $\lambda = 0$, while achieving the highest average accuracy at $\lambda = 0.5$.

**The Location of FFM.** We evaluate the approach by inserting FFM into the 3rd, 6th, and 9th layers of the network. Additionally, we evaluate the post-fusion method by positioning FFM after the ViT feature encoding. As shown in Table 10, the pre-fusion method proves to be the most effective. We deem that high-level semantic representation may already lose most of the fine-grained perception and alignment characteristics, demonstrating the limitation of current methods that use the high-level semantic fusion strategy.

*Table 10.* **Performance analysis under different locations for FFM.** We show the best score in bold.

| Location | $m$AP | R-1 | R-5 | R-10 | Average |
|---|---|---|---|---|---|
| 0(Before ViT) | **80.7** | **83.5** | **91.9** | **94.1** | **87.5** |
| 3 | 50.6 | 50.5 | 62.9 | 70.7 | 58.7 |
| 6 | 74.0 | 78.7 | 86.8 | 91.1 | 82.6 |
| 9 | 75.3 | 78.3 | 86.8 | 90.1 | 82.6 |
| 12(After ViT) | 76.7 | 79.7 | 86.8 | 92.0 | 83.8 |

## 5. Conclusion

In this paper, we propose **MFRNet**, a novel framework for multi-modal object ReID that enhances feature fusion and representation through the mixture of experts (MoE). Specifically, we introduce a multi-modal feature fusion module (FFM) to enable fine-grained cross-modal interaction. Following this, the designed feature representation module (FRM) extracts modality-specific and modality-shared features through a parameter-efficient expert architecture. Extensive experiments on the RGBNT201, RGBNT100, and MSVR310 datasets demonstrate the effectiveness and efficiency of our model.

## Impact Statement

This paper presents work whose goal is to advance the field of Machine Learning. There are many potential societal consequences of our work, none of which we feel must be specifically highlighted here.

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
