# OpenReview forum: "Multi-Modal Object Re-identification via Sparse Mixture-of-Experts"
_ICML.cc/2025/Conference — ICML 2025 poster_

### Official Review · Reviewer_am8H · 2025-03-06

**Overall Recommendation:** 4

**Summary:**

This work introduces MFRNet, which mitigates insufficient interaction and feature imbalance via two modules. The Feature Fusion Module (FFM) uses a mixture-of-generators for pixel-level alignment, while the Feature Representation Module (FRM) employs a mixture-of-experts for balanced modality-shared and modality-specific features.

**Claims And Evidence:**

This work presents two key claims. The first claim highlights the limitations of recent approaches in modal interaction. To address this, the authors advocate for pixel-level interactions instead of feature-level interactions. Given the characteristics of multimodal images, this proposition is technically sound, as further validated by subsequent experiments. The second claim concerns the trade-off between feature representation quality and computational efficiency in existing methods. To tackle this issue, the authors propose leveraging a Mixture of Experts (MoE) to enable dynamic parameter allocation and streamline the model structure. This paradigm is widely recognized as effective, as MoE not only adapts to data variations but also enhances model efficiency by reducing redundancy. Therefore, the claims in this paper are well-founded.

**Essential References Not Discussed:**

This paper could be better by considering published literature in the cross-modality ReID community such as [1, 2].

[1] Zhang, Yukang, and Hanzi Wang. "Diverse embedding expansion network and low-light cross-modality benchmark for visible-infrared person re-identification." Proceedings of the IEEE/CVF conference on computer vision and pattern recognition. 2023.

[2] Park, Hyunjong, et al. "Learning by aligning: Visible-infrared person re-identification using cross-modal correspondences." Proceedings of the IEEE/CVF international conference on computer vision. 2021.

**Experimental Designs Or Analyses:**

The experiments in this paper utilize Cumulative Matching Characteristics (CMC) and mean Average Precision (mAP) as evaluation metrics. The experimental setup includes comparative experiments (Table 1, 2, 3), ablation studies (Table 4), hyperparameter configurations (Table 5, 6, 7, 8, 9), and visualization (Figure 3). While the experiments are comprehensive, the analysis lacks depth. For instance, Table 3 presents results for M(TIR) without providing a corresponding discussion, in which the performance of MFRNet is lower than TOPReID.

**Methods And Evaluation Criteria:**

The paper presented in this paper primarily consists of two key modules: the Feature Fusion Module (FFM) and the Feature Representation Module (FRM). The FFM replaces traditional feature-level interactions with fine-grained interactions, which is particularly suitable given the unique characteristics of multispectral images. Meanwhile, the FRM leverages a Mixture of Experts (MoE) framework to balance modality-specific feature representation and structural redundancy, which is also technically sound.

**Other Comments Or Suggestions:**

-The notation for Mean and Std in Equations (12) and (13) is inconsistent in font style. Please ensure uniform formatting for clarity and consistency.

- In Equations (5) and (9), $W_1$ and $W_9$ originate from different modules and should be explicitly distinguished to avoid confusion

**Other Strengths And Weaknesses:**

Most of the strengths and limitations have been discussed in the previous sections. Overall, the paper is well-structured, with a clear research motivation and methodology. It effectively addresses the stated problem and achieves strong performance. However, a key limitation is that the implementation of the Mixture of Experts (MoE) strategy is simple, lacks significant technical innovation.

**Questions For Authors:**

Besides the above mentions, I have one more question: While the authors' claim is convincing, the results in Table 7 do not provide sufficient evidence that the Feature Fusion Module (FFM) requires more experts. It would be beneficial to include a comparison with the case where the number of experts in FFM is set to 1 to better validate this claim.

**Relation To Broader Scientific Literature:**

The MFRNet in this paper is clearly presented and of good quality. The proposed two modules are built upon the Mixture of Experts (MoE) framework, which is not a novel technique. While MoE has been adopted in several topics existing in the computer vision community, its application to multimodal person re-identification, particularly from both interaction and representation perspectives, still holds meaning.

**Theoretical Claims:**

This paper does not include extensive theoretical proofs; however, there are some ambiguities in the use of mathematical notations. For instance, in Equations (5) and (9), $W_1$ and $W_9$ originate from different modules and should be explicitly distinguished to avoid confusion.

---

> ### Author Rebuttal · Authors · 2025-03-31
>
> > **Q1:** While the experiments are comprehensive, the analysis lacks depth. For instance, Table 3 presents results for M(TIR) without providing a corresponding discussion in which the performance of MFRNet is lower than TOPReID.
> >
> **A1:**  Thank you for your suggestion. TOP-ReID benefits from specific training tailored for handling modality-missing scenarios, whereas our method does not include dedicated training for such cases. During training, we utilize all modalities. For testing, when a modality is missing, we supplement it using Equation 6 from the paper. For example, if RGB is missing, its features are generated based on NIR and TIR as follows: $I_R=w_R^{N}(I_N) \times g(I_N) + w_R^{T}(I_T) \times g(I_T)$. Here, $g(I_N)$ represents the features for RGB generated from NIR, $g(I_T)$ denotes the features for RGB generated from TIR, $w_R^{N}(I_N)$ and $w_R^{T}(I_T)$ are the respective weights.) To ensure model generalizability, we do not employ dedicated training methods for handling missing modalities. Notably, performing specific missing modality training could yield even better results. As shown in Table A, training our model to address the missing TIR led to a 7.7% improvement in mAP and a 7.0% increase in R-1 compared to TOP-ReID.
>
> Table A: Experimental results for missing TIR modality.
>
> |  | mAP | R-1 | R-5 | R-10 |
> | --- | --- | --- | --- | --- |
> | TOP-ReID | 51.9 | 54.5 | - | - |
> | Ours | 51.6 | 49.5 | 67.7 | 76.7 |
> | Ours (RGB+NIR) | **59.6** | **61.5** | **72.7** | **80.6** |
>
> > **Q2:** This paper could be better by considering published literature in the cross-modality ReID community such as [1, 2]. [1] Zhang, Yukang, and Hanzi Wang. "Diverse embedding expansion network and low-light cross-modality benchmark for visible-infrared person re-identification." Proceedings of the IEEE/CVF conference on computer vision and pattern recognition. 2023. [2] Park, Hyunjong, et al. "Learning by aligning: Visible-infrared person re-identification using cross-modal correspondences." Proceedings of the IEEE/CVF international conference on computer vision. 2021.
> >
>
> **A2:** Thanks for your suggestion. We have included these papers in Sec. 2 to make our work more comprehensive. If you have any other recommended works that are related to this paper, you can provide them in the discussion. We are willing to add them in.
>
> > **Q3:** Besides the above mentions, I have one more question: While the authors' claim is convincing, the results in Table 7 do not provide sufficient evidence that the Feature Fusion Module (FFM) requires more experts. It would be beneficial to include a comparison with the case where the number of experts in FFM is set to 1 to better validate this claim.
> >
> **A3:**  Thank you for your suggestion. We conducted additional experiments as shown in Table B. The results indicate that the optimal performance, with an average accuracy of 87.5%, is achieved when the number of experts in the FFM is set to 3.
>
> Table B: Performance analysis under different expert numbers for FFM.
>
> | Number | mAP | R-1 | R-5 | R-10 | Average |
> | --- | --- | --- | --- | --- | --- |
> | 1 | 78.7 | 81.0 | 90.8 | 93.5 | 86.0 |
> | 2 | 74.9 | 78.6 | 86.8 | 90.8 | 82.8 |
> | **3** | **80.7** | **83.5** | **91.9** | **94.1** | **87.5** |
> | 6 | 76.9 | 80.4 | 88.2 | 90.6 | 84.0 |
> | 9 | 79.2 | 82.3 | 90.7 | 93.5 | 86.4 |
>
> > **Q4:** In Equations (5) and (9),  $W_1$ and $W_9$ originate from different modules and should be explicitly distinguished to avoid confusion.
> >
> **A4:**  Thank you for your suggestion. We have added markers in the top-right corner to distinguish between different components.
>
> > **Q5:**  The notation for Mean and Std in Equations (12) and (13) is inconsistent in font style. Please ensure uniform formatting for clarity and consistency.
> >
> **A5:**  Thank you for your suggestion. We have corrected them following your comments.

---

### Official Review · Reviewer_yVLW · 2025-03-07

**Overall Recommendation:** 4

**Summary:**

This paper introduces MFRNet for multi-modal object re-identification. This approach addresses two core issues: insufficient pixel-level feature interaction and difficulty balancing between shared and specific modality features. The proposed Feature Fusion Module (FFM) fosters fine-grained cross-modal interaction, while the Feature Representation Module (FRM) efficiently merges modality-shared and modality-specific representations in a unified network. Experiments on three public datasets show that MFRNet significantly improves both accuracy and efficiency, with minimal computational overhead.

**Claims And Evidence:**

This work‘s claims are intuitive, convincing, and supported by its experiments.

**Essential References Not Discussed:**

The necessary references have been discussed, but the related work section is somewhat lengthy. It would be better to condense it.

**Experimental Designs Or Analyses:**

The experimental design of the paper is comprehensive. In addition to the ablation studies on FFM and FRM, the paper also provides a detailed discussion of relevant hyperparameters, making the overall experiments convincing.

**Methods And Evaluation Criteria:**

It looks well. These two limitations addressed in this topic are reasonable, the introduction of FFM and FRM is suitable for this task.

**Other Comments Or Suggestions:**

I have no more comments, please check the aforementioned parts.

**Other Strengths And Weaknesses:**

Strengths：
- The proposed framework is well-structured, with clearly defined motivations and a logically designed modular architecture. The integration of Feature Fusion Module (FFM) and Feature Representation Module (FRM) effectively enhances feature interaction and representation learning, addressing key challenges in multi-modal object re-identification.
- The performance of this work is quite strong, achieving a significant improvement over the previous state-of-the-art.

Weaknesses:
- The captions lack some details. What do ‘RE’ and ‘GE’ mean in Figure 2?
- Eq.12 and Eq.13 is not alignment.

**Questions For Authors:**

1. Given that prior works such as TOP-ReID [1], EDITOR [2], and RSCNet [3] both use an ImageNet-based ViT as the backbone, while MFRNet adopts a CLIP-based ViT, I am curious about MFRNet’s performance when using an ImageNet-based ViT.

2. Some details of the testing phase are not entirely clear. In Table 3, I would like to know how the FFM work in the absence of certain modalities.

3. Some captions lack some details. What do ‘RE’ and ‘GE’ mean in Figure 2?

[1] "Top-reid: Multi-spectral object re-identification with token permutation." AAAI 2024.

[2] "Magic tokens: Select diverse tokens for multi-modal object re-identification." CVPR 2024

[3] "Representation Selective Coupling via Token Sparsification for Multi-Spectral Object Re-Identification." IEEE Transactions on CSVT (2024).

**Relation To Broader Scientific Literature:**

Adapting to multi-modal data using MoE is an appropriate approach and is widely employed in current Multimodal Large Language Models. This work integrates this technique into the task of multi-modal object re-identification. Therefore, the impact of this paper is moderate, but it provides some valuable insights into methods in this field.

**Theoretical Claims:**

I have reviewed the methodology and corresponding equations in this work, and they appear to be both reliable and reasonable. However, Figure 2 may be misleading. Specifically, in Section 3.1, the description of the FFM states that transformations for all three modalities occur simultaneously. Yet, Figure 2 only illustrates interactions involving the RGB image, which may not fully represent the fusion process.

---

> ### Author Rebuttal · Authors · 2025-03-31
>
> > **Q1:** Specifically, in Section 3.1, the description of the FFM states that transformations for all three modalities occur simultaneously. Yet, Figure 2 only illustrates interactions involving the RGB image, which may not fully represent the fusion process.
> >
> **A1:**  Thank you for your suggestion. This transformation process applies equally to all three modalities. While the figure specifically illustrates the NIR+RGB->TIR interaction, we have conducted the same interaction for NIR and RGB. We will revise the caption of Figure 2 in light of your feedback to ensure the process is described clearly.
>
> > **Q2:** The necessary references have been discussed, but the related work section is somewhat lengthy. It would be better to condense it.
> >
> **A2:**  Thank you for your suggestion. We will accordingly reduce the content of the related work section.
>
> > **Q3:** Given that prior works such as TOP-ReID [1], EDITOR [2], and RSCNet [3] both use an ImageNet-based ViT as the backbone, while MFRNet adopts a CLIP-based ViT, I am curious about MFRNet’s performance when using an ImageNet-based ViT.
> >
> **A3:**  Following your suggestions, we conducted experiments on our server to evaluate TOP-ReID and our method using ImageNet-based ViT and CLIP-based ViT. As shown in Table A, the results demonstrate that both methods perform better with CLIP-based ViT compared to ImageNet-based ViT. Specifically, with ImageNet-based ViT, our method outperforms TOP-ReID by approximately 2.2% in mAP and 3.7% in R1. With CLIP-based ViT, our method surpasses TOP-ReID by about 9.7% in mAP and 9.6% in R1. These findings underscore the superior performance of our method over TOP-ReID under identical settings and backbones.
>
> Table A: Performance comparison on ImageNet-based and CLIP-based ViT.
>
> | Methods | mAP | R-1 | R-5 | R-10 |
> | --- | --- | --- | --- | --- |
> | ViT: TOP-ReID | 67.4 | 69.1 | 80.9 | 86.0 |
> | ViT: Ours | **69.6** | **72.8** | **84.9** | **90.6** |
> | CLIP-ViT: TOP-ReID | 71.0 | 73.9 | 81.6 | 86.6 |
> | CLIP-ViT: Ours | **80.7** | **83.5** | **91.9** | **94.1** |
>
> > **Q4:** Some details of the testing phase are not entirely clear. In Table 3, I would like to know how the FFM work in the absence of certain modalities.
> >
> **A4:**  Following your suggestion, we will refine this section in future versions to enhance clarity. For the missing modality, as illustrated in Equation 6 (lines 190-199), the other two modalities are utilized for prediction. For example, if RGB is missing, its features are generated based on NIR and TIR as follows: $I_R=w_R^{N}(I_N) \times g(I_N) + w_R^{T}(I_T) \times g(I_T)$. Here, $g(I_N)$ denotes the features for RGB generated from NIR, $g(I_T)$ represents the features for RGB generated from TIR, $w_R^{N}(I_N)$ and $w_R^{T}(I_T)$ are the respective weights.
>
> > **Q5:** Some captions lack some details. What do ‘RE’ and ‘GE’ mean in Figure 2?
> >
> **A5:**  We sincerely apologize for any inconvenience caused. 'GE' refers to the generation experts in FFR, while 'RE' represents the representation experts in FRM. Based on your suggestion, we will revise Figure 2 accordingly.
>
> > **Q6:** Eq.12 and Eq.13 are not aligned.
> >
> **A6:**  Thank you for your suggestion. We have correctted it in these days.

---

### Official Review · Reviewer_hHKf · 2025-03-10

**Overall Recommendation:** 4

**Summary:**

This paper proposes a novel Multi-modal Fusion and Representation Network (MFRNet) approach for multi-modal object re-identification, inspired by the sparse Mixture-of-Experts (MoE) paradigm. The proposed framework enhances performance by introducing a Feature Fusion Module (FFM) for fine-grained pixel-level cross-modal interaction and a Feature Representation Module (FRM) to extract modality-shared and modality-specific features dynamically. Experimental evaluations on three benchmark datasets, RGBNT201, RGBNT100, and MSVR310, demonstrate that the proposed method achieves superior performance compared to existing state-of-the-art methods.

**Claims And Evidence:**

Yes, the claims in the submission are generally supported by clear and convincing evidence. The authors present two well-structured modules with empirical results that support the proposed modules.

**Essential References Not Discussed:**

Yes, related works that are essential to understanding the key contributions of the paper are adequately cited and discussed.

**Experimental Designs Or Analyses:**

Yes, I reviewed the experimental design and related analyses and found it to be sound and sufficient.

**Methods And Evaluation Criteria:**

Yes, the proposed methods and evaluation criteria make sense for addressing the problem. The selected benchmark datasets are appropriate, and the evaluation approach is well-aligned with the research objectives.

**Other Comments Or Suggestions:**

1. Page 2, Line 127-128: ‘we propose MRFNet’ should be ‘we propose MFRNet’.

2. Page 4, Line 217-218: ‘we aim to modality-specific’ should be ‘we aim to capture modality-specific’?

3. Figure 2 is slightly cluttered, and the author seems to have forgotten to describe GE and RE.

**Other Strengths And Weaknesses:**

Strengths:

1. The paper proposes an effective sparse Mixture-of-Experts (MoE) architecture for multi-modal ReID, achieving notable performance improvements over existing state-of-the-art models.

2. The paper conducts comprehensive ablation studies, clearly demonstrating the contribution and effectiveness of each module.

3. MFRNet achieves significant performance gains on three public datasets, validating the effectiveness of the proposed approach.

4. MFRNet exhibits a certain level of robustness in scenarios with missing modalities, even without explicit training for such cases.

Weaknesses:

1. While the MFRNet presents a novel perspective within multi-modal ReID by introducing a sparse Mixture-of-Experts (MoE) framework, its overall novelty remains moderate. It is more like a representing of an effective combination of existing techniques.

**Questions For Authors:**

1. The visualization section appears to focus solely on the Feature Representation Module, while the visualization of the Feature Fusion Module is also essential for a comprehensive analysis.

2. Can the authors provide a more detailed comparison of computations with different methods?

3. The author mentioned that ‘TOP-ReID has a specific training phase for the modality missing condition’ while MFRNet does not. Will MFRNet be better than TOP-ReID in the 'M (TIR)' protocol with the same specific training phase?

4. Why can the Feature Fusion Module reduce computational complexity?

**Relation To Broader Scientific Literature:**

The key contributions of this paper relate closely to the broader scientific literature in the following the Sparse Mixture-of-Experts (MoE) Integration. The paper builds upon the concept of Sparse MoE, which has been widely explored in deep learning to improve model expressiveness and parameter efficiency. Specifically, prior works such as GMoE [1] have leveraged MoE frameworks in domain generalization problem. The current paper extends this idea explicitly to multi-modal object re-identification, demonstrating its effectiveness in this context. Overall, the proposed contributions build upon and advance existing findings in sparse MoE modeling and achieve fine-grained pixel-level interactions, and multi-modal representation balancing, extending these concepts specifically into the multi-modal object re-identification task and demonstrating clear empirical advantages over prior state-of-the-art approaches.

Reference:

[1] Li, Bo, et al. "Sparse Mixture-of-Experts are Domain Generalizable Learners." In The Eleventh International Conference on Learning Representations, 2023.

**Theoretical Claims:**

Yes, I have checked the correctness of the proofs for the theoretical claims and found no issues. The formulations and proofs are well-founded and align with standard methodologies in the field.

---

> ### Author Rebuttal · Authors · 2025-03-31
>
> > **Q1:** While the MFRNet presents a novel perspective within multi-modal ReID by introducing a sparse Mixture-of-Experts (MoE) framework, its overall novelty remains moderate. It is more like a representing of an effective combination of existing techniques.
> >
> **A1:** Our novelty primarily lies in two aspects: generation and representation. For FFM generation, unlike previous methods that rely on coarse-grained feature interactions, we creatively harness the pixel-level consistency of multimodal data (pixel-wise alignment) to achieve fine-grained multimodal interaction. For FRM representation, our approach avoids the independent encoding of each modality. Instead, we introduce multimodal feature representation experts based on the MoE architecture, achieving a balance between modality-specific and shared features while maintaining minimal computational cost.
>
> > **Q2:** Figure 2 is slightly cluttered, and the author seems to have forgotten to describe GE and RE.
> >
> **A2:**  We sincerely apologize for any inconvenience caused. GE refers to the generation experts in FFR, while RE represents the representation experts in FRM. Based on your suggestion, we will revise Figure 2 accordingly.
>
> > **Q3:** The visualization section appears to focus solely on the Feature Representation Module, while the visualization of the Feature Fusion Module is also essential for a comprehensive analysis.
> >
> **A3:**  Thank you for your suggestion. In response, we have also visualized FFM. However, due to the constraints of the rebuttal format, we are unable to include the image directly. Nonetheless, it will be incorporated in future versions of the paper.
>
> > **Q4:** Can the authors provide a more detailed comparison of computations with different methods?
> >
> **A4:**  Following your suggestion, we compared the computational costs with three recent works. As shown in Table B, our method not only demonstrates significant improvements in metrics such as mAP and R-1 but also achieves the lowest Params and FLOPs, with values of 57.1M and 22.1G, respectively.
>
> Table A: Comparison of computational cost with recent methods. The best results are shown in bold.
>
> |  | mAP | R-1 | R-5 | R-10 | Params(M) | Flops(G) |
> | --- | --- | --- | --- | --- | --- | --- |
> | HTT | 71.1 | 73.4 | 83.1 | 87.3 | 85.6 | 33.1 |
> | EDITOR | 66.5 | 68.3 | 81.1 | 88.2 | 117.5 | 38.6 |
> | TOP-ReID | 72.3 | 76.6 | 84.7 | 89.4 | 278.2 | 34.5 |
> | Ours | **80.7** | **83.5** | **91.9** | **94.1** | **57.1** | **22.1** |
>
> > **Q5:** The author mentioned that ‘TOP-ReID has a specific training phase for the modality missing condition’ while MFRNet does not. Will MFRNet be better than TOP-ReID in the 'M (TIR)' protocol with the same specific training phase?
> >
> **A5:**  Following your suggestion, we conducted additional experiments targeting the 'M (TIR)' protocol. As shown in Table B, training our model to address the missing of TIR resulted in a 7.7% improvement in mAP and a 7.0% increase in R-1 compared to TOP-ReID.
>
> Table B: Experimental results for missing TIR modality.
>
> | M (TIR) | mAP | R-1 | R-5 | R-10 |
> | --- | --- | --- | --- | --- |
> | TOP-ReID | 51.9 | 54.5 | - | - |
> | Ours | 51.6 | 49.5 | 67.7 | 76.7 |
> | Ours (RGB+NIR) | **59.6** | **61.5** | **72.7** | **80.6** |
>
> > **Q6:** Why can the FRM reduce computational complexity?
> >
> **A6:**  Thank you for your suggestion. As demonstrated in Equation 8, the FRM module leverages RepAdapter to build MoE representations. Compared to a traditional MLP, RepAdapter reduces computational costs by incorporating two convolutional layers. If replaced with the original MLP structure, as indicated in Table C, the parameters and FLOPs would increase by 3.7M (from 53.5 to 57.2) and 1.4G (from 20.7 to 22.1), respectively.
>
> Table C: Comparison of FRM and original MLP.
>
> |  | Params (M) | FLOPs (G) |
> | --- | --- | --- |
> | Original MLP | 57.2 | 22.1 |
> | FRM | 53.5 | 20.7 |
>
> > **Q7:** There are several typos in the paper.
> >
> **A7:** Thank you for your suggestion. We will correct the typos in the updated version of the paper.

---

> > ### Comment · Reviewer_hHKf · 2025-04-02
> >
> > Thank you for your response. My previous concerns have been well addressed. The paper demonstrates strong performance and efficiency, and I’m willing to raise my score to Accept. I hope the authors will include these additional experiments in the final version, as they would be highly valuable.

---

### Official Review · Reviewer_PMp6 · 2025-03-10

**Overall Recommendation:** 4

**Summary:**

This work presents the Modality Fusion and Representation Network (MFRNet) aiming to address the limitations in modality interaction and representation of recent works. Two modules named Feature Fusion Module (FFM) and Feature Representation Module (FRM) are proposed to tackle the interaction and representation limitations, respectively. Both two modules follow a MoE structure. The FFM employs multiple generator experts to adaptively provide fine-grained interaction information, while the FRM employs diverse representation experts to extract and combine modality-specific and modality-shared features. Experiments are conducted in multi-modal ReID datasets. Both qualitative and quantitative results are reported to show the effectiveness of each component in the method.

**Claims And Evidence:**

This paper proposes two modules to solve two claimed problems. From its motivation and implementation, the modules present in this paper effectively support its claims. Experiments and visualization result also show the improvements in these two modules.

**Essential References Not Discussed:**

I think essential references are adequate discussed in this paper.

**Experimental Designs Or Analyses:**

I have checked the experiments of this work. Most experiments are appropriate and complete. While there still exist several concerns:

1. From Table 4, we can observe that the FRM has decreased the 'Params' and 'FLOPs'. Generally, the MoE structure should keep similar or slightly higher 'Params' and 'FLOPs' with the baseline during the inference.

2. The discussion of 'Params' and 'FLOPs' only contains this method itself. I understand that it can be similar efficiency to the baseline method. But it would be better if the author could show a comparison with recent methods.

3. Table 7 may not enough to show that 3 is the optimal selection.

4. FFM doesn't seem to conflict with VIT. It seems can be inserted in the network, but it is not discussed at all.

**Methods And Evaluation Criteria:**

Yes, the problem of multi-modal person re-identification is important and valuable when considering real application scenarios. The proposed method makes sense in this topic due to its performance and efficiency.

**Other Comments Or Suggestions:**

This is a paper with sufficient motivation and an interesting solution. However, as mentioned before, there are several issues that limit the value of this work. If the two weaknesses can be addressed, I would like to improve my score.

**Other Strengths And Weaknesses:**

Strengths：
1. The idea of using MoE in interaction and representation is somewhat interesting in this topic.
2. This method has achieved strong performance while maintaining good efficiency.

Weaknesses:
1. There still several typos:
- The caption in Table 6, ‘…for FRM’ -> ‘…for FRM’
- The caption of Table 3 and Table 4 should be bolded.

**Questions For Authors:**

1. From Table 4, we can observe that the FRM has decreased the 'Params' and 'FLOPs'. However, as an MoE structure, even ignoring the router's parameter, the 'Params' and 'FLOPs' should remain the same with the baseline method. How it can even decrease the 'Params' and 'FLOPs'?
2. The discussion of 'Params' and 'FLOPs' only contains this method itself. I understand that it can be similar efficiency to the baseline method. But, can you show the comparison with other methods?
3. From Table 7, what is the performance when number of experts of FFM is less than 3? Table 7 is insufficient to indicate that 3 is optimal.
4. FFM doesn't seem to conflict with VIT. Why is Table 9 only discussing before and after ViT?

**Relation To Broader Scientific Literature:**

This work uses the concept of MoE to solve the limitations of recent works in two ways. Compared to recent methods such as TOP-ReID [1] which uses multiple networks for different modalities and interacts with them in the final, this work largely simplified the network structure while maintaining the dynamical specific encoding processing and achieving fine-grained for each modality. Because this work aims to solve the specific limitations of this topic, I think it is moderately related to the general re-identification task.
[1] Wang, Yuhao, et al. "Top-reid: Multi-spectral object re-identification with token permutation." Proceedings of the AAAI Conference on Artificial Intelligence. Vol. 38. No. 6. 2024.

**Theoretical Claims:**

This work proposes two modules that are somewhat reasonable from both its claims and design. The FFM uses the MoE structure for the image completion, while the FRM uses the MoE structure for the representation. Since the MoE structure has already shown its ability in LLM to adapt to multiple modality data, using such a structure to strengthen the ability of multi-modal re-identification reasonable.

---

> ### Author Rebuttal · Authors · 2025-03-31
>
> ## Response to Reviewer PMp6:
>
> > **Q1:** From Table 4, we can observe that the FRM has decreased the 'Params' and 'FLOPs'. Generally, the MoE structure should keep similar or slightly higher 'Params' and 'FLOPs' with the baseline during the inference.
> >
> **A1:** Thank you for your suggestion. As demonstrated in Equation 8, the FRM module leverages RepAdapter to build MoE representations. Compared to a traditional MLP, RepAdapter reduces computational costs by incorporating two convolutional layers. If replaced with the original MLP structure, as indicated in Table A, the parameters and FLOPs would increase by 3.7M (from 53.5 to 57.2) and 1.4G (from 20.7 to 22.1), respectively.
>
> Table A: Comparison of FRM and original MLP.
>
> |  |  Params (M) |  FLOPs (G) |
> | --- | --- | --- |
> | Original MLP | 57.2 | 22.1 |
> | FRM | 53.5 | 20.7 |
>
> > **Q2:** The discussion of 'Params' and 'FLOPs' only contains this method itself. I understand that it can be similar efficiency to the baseline method. But it would be better if the author could show a comparison with recent methods.
>
> **A2:** Following your suggestion, we compared the computational costs with three recent works. As shown in Table B, our method not only demonstrates significant improvements in metrics such as mAP and R-1 but also achieves the lowest Params and FLOPs, with values of 57.1M and 22.1G, respectively.
>
> Table B: Comparison of computational cost with recent methods.
>
> |  | mAP | R-1 | R-5 | R-10 | Params(M) | Flops(G) |
> | --- | --- | --- | --- | --- | --- | --- |
> | HTT | 71.1 | 73.4 | 83.1 | 87.3 | 85.6 | 33.1 |
> | EDITOR | 66.5 | 68.3 | 81.1 | 88.2 | 117.5 | 38.6 |
> | TOP-ReID | 72.3 | 76.6 | 84.7 | 89.4 | 278.2 | 34.5 |
> | Ours | **80.7** | **83.5** | **91.9** | **94.1** | **57.1** | **22.1** |
>
> > **Q3:** Table 7 may not enough to show that 3 is the optimal selection.
> >
> **A3:** Thank you for your suggestion. We conducted additional experiments as shown in Table C. The results indicate that the optimal performance, with an average accuracy of 87.5%, is achieved when the number of experts in the FFM is set to 3.
>
> Table C: Performance analysis under different expert numbers for FFM.
>
> | Number | mAP | R-1 | R-5 | R-10 | Average |
> | --- | --- | --- | --- | --- | --- |
> | 1 | 78.7 | 81.0 | 90.8 | 93.5 | 86.0 |
> | 2 | 74.9 | 78.6 | 86.8 | 90.8 | 82.8 |
> | 3 | **80.7** | **83.5** | **91.9** | **94.1** | **87.5** |
> | 6 | 76.9 | 80.4 | 88.2 | 90.6 | 84.0 |
> | 9 | 79.2 | 82.3 | 90.7 | 93.5 | 86.4 |
>
> > **Q4:** FFM doesn't seem to conflict with VIT. It seems can be inserted in the network, but it is not discussed at all.
> >
> **A4:** Following your suggestion, we further validated the approach by inserting FFM into the 3rd, 6th, and 9th layers of the network. As shown in Table D, applying FFM before the ViT leverages the pixel-by-pixel alignment of multimodal image data more effectively, resulting in optimal model performance.
>
> Table D: Performance analysis under different locations for FFM.
>
> | Location | mAP | R-1 | R-5 | R-10 | Average |
> | --- | --- | --- | --- | --- | --- |
> | 0 (Before ViT) | **80.7** | **83.5** | **91.9** | **94.1** | **87.5** |
> | 3 | 50.6 | 50.5 | 62.9 | 70.7 | 58.7 |
> | 6 | 74.0 | 78.7 | 86.8 | 91.1 | 82.6 |
> | 9 | 75.3 | 78.3 | 86.8 | 90.1 | 82.6 |
> | 12 (After ViT) | 76.7 | 79.7 | 86.8 | 92.0 | 83.8 |
>
> > **Q5:** There are still several typos in the paper.
> >
> **A5:** Thank you for your suggestion. We will correct these typos in the updated version of the paper.

---

> > ### Comment · Reviewer_PMp6 · 2025-04-03
> >
> > Thank you for your response and a well-written rebuttal. Most of my concerns have been effectively addressed. However, there are still a few points that require further clarification. I will decide my final score on the answers to these remaining questions:
> >
> > 1. The authors clarified that the reduction in parameters brought by FRM mainly stems from structure changes. Therefore, I believe it is necessary to also compare how much performance improvement is achieved by these structural modifications. This would help clarify the necessity of the MoE structure within FRM.
> >
> > 2. The performance curve shown in Table C is rather unusual, resembling a W-shape. It remains unclear whether there are additional performance peaks beyond the value of 9.

---

> > > ### Author Response · Authors · 2025-04-04
> > >
> > > > **Q1:** The authors clarified that the reduction in parameters brought by FRM mainly stems from structure changes. Therefore, I believe it is necessary to also compare how much performance improvement is achieved by these structural modifications. This would help clarify the necessity of the MoE structure within FRM.
> > > >
> > > **A1:** Thank you for your suggestion. To better illustrate the significance of the MoE structure in FRM, we remove the MoE while retaining other components. As shown in Table E, using only RepAdapter (excluding MoE) improves the average performance by 3.1%. Adding the MoE further boosts performance, resulting in a 3.5% increase compared to its absence, highlighting the MoE's positive impact on model performance.
> > >
> > > Table E: Performance analysis of FRM (RepAdapter+MoE).
> > >
> > > | Method | mAP | R-1 | Average |
> > > | --- | --- | --- | --- |
> > > | Base | 69.2 | 76.3 | 72.7 |
> > > | +RepAdapter | 74.2 | 77.5 | 75.8 |
> > > | +RepAdapter+MoE (FRM ) | **77.8** |  **80.9** | **79.3** |
> > >
> > > > **Q2:** The performance curve shown in Table C is rather unusual, resembling a W-shape. It remains unclear whether there are additional performance peaks beyond the value of 9.
> > > >
> > > **A2:**  Following your suggestion, we conduct additional experiments with parameters exceeding 9. As illustrated in the revised Table C, model performance decreases when the number of experts surpasses 9. Overall, the optimal performance is achieved when the number of experts is set to 3.
> > >
> > > Table C: Performance analysis under different expert numbers for FFM.
> > >
> > > | Number | mAP | R-1 | R-5 | R-10 | Average |
> > > | --- | --- | --- | --- | --- | --- |
> > > | 1 | 78.7 | 81.0 | 90.8 | 93.5 | 86.0 |
> > > | 2 | 74.9 | 78.6 | 86.8 | 90.8 | 82.8 |
> > > | 3 | **80.7** | **83.5** | **91.9** | **94.1** | **87.5** |
> > > | 6 | 76.9 | 80.4 | 88.2 | 90.6 | 84.0 |
> > > | 9 | 79.2 | 82.3 | 90.7 | 93.5 | 86.4 |
> > > | 10 | 76.4 | 80.1 | 88.0 | 92.0 | 84.1 |
> > > | 11 | 74.1 | 78.9 | 87.2 | 92.2 | 83.1 |
> > > | 12 | 74.1 | 77.8 | 86.8 | 91.1 | 82.4 |
> > > | 15 | 74.4 | 77.9 | 87.0 | 90.4 | 82.4 |

---

### Decision · Program_Chairs · 2025-05-01

**Decision:**

Accept (poster)

**Comment:**

This paper received four positive ratings, with all reviewers generally inclined to accept it. It presents a Modality Fusion and Representation Network (MFRNet) for multi-modal object re-identification, which aims to address the two challenges of insufficient pixel-level feature interaction and the difficult balance between shared and modality-specific features. According to the reviews, the paper is well-organized with a clear motivation. It provides a comprehensive ablation study demonstrating the contribution and effectiveness of each module. In particular, the experiments show that MFRNet still exhibits a certain degree of robustness to the scenario where the missing mode is not explicitly trained.
The authors have addressed the concerns raised, resolving most of the doubts. Overall, it is a good work, and the Area Chair (AC) recommends accepting the paper.